# *In vitro* activity of aryl-thiazole derivatives against *Schistosoma mansoni* schistosomula and adult worms

**Adriana S. A. Pereira**[1,2]**, Gilbert O. Silveira**[1,2]**, Murilo S. Amaral**[1]**, Sinara M. V. Almeida**[3,4]**, Jamerson F. Oliveira**[3]**, Maria C. A. Lima**[3]**, Sergio Verjovski-Almeida**[1,2]*****

**1** Instituto Butantan, São Paulo, Brasil, **2** Departamento de Bioquímica, Instituto de Química, Universidade de São Paulo, São Paulo, Brasil, **3** Universidade Federal de Pernambuco, Departamento de Antibióticos, Recife, Pernambuco, Brasil, **4** Universidade de Pernambuco, Campus Garanhuns, Garanhuns, Pernambuco, Brasil

\* verjo@iq.usp.br

**Data Availability Statement:** All relevant data are within the paper and its Supporting Information files.

## Abstract

Schistosomiasis is caused by a trematode of the genus *Schistosoma* and affects over 200 million people worldwide. The only drug recommended by the World Health Organization for treatment and control of schistosomiasis is praziquantel. Development of new drugs is therefore of great importance. Thiazoles are regarded as privileged structures with a broad spectrum of activities and are potential sources of new drug prototypes, since they can act through interactions with DNA and inhibition of DNA synthesis. In this context, we report the synthesis of a series of thiazole derivatives and their *in vitro* schistosomicidal activity by testing eight molecules (NJ03-08; NJ11-12) containing thiazole structures. Parameters such as motility and mortality, egg laying, pairing and parasite viability by ATP quantification, which were influenced by these compounds, were evaluated during the assays. Scanning electron microscopy (SEM) was utilized for evaluation of morphological changes in the tegument. Schistosomula and adult worms were treated *in vitro* with different concentrations (6.25 to 50 μM) of the thiazoles for up to 5 and 3 days, respectively. After *in vitro* treatment for five days with 6.25 μM NJ05 or NJ07 separately, we observed a decrease of 30% in schistosomula viability, whilst treatment with NJ05+NJ07 lead to a reduction of 75% in viability measured by ATP quantitation and propidium iodide labeling. Adult worms' treatment with 50 μM NJ05, NJ07 or NJ05 + NJ07 showed decreased motility to 30–50% compared with controls. Compound NJ05 was more effective than NJ07, and adult worm viability after three days was reduced to 25% in parasites treated with 50 μM NJ05, compared with a viability reduction to 40% with 50 μM NJ07. SEM analysis showed severe alterations in adult worms with formation of bulges and blisters throughout the dorsal region of parasites treated with NJ05 or NJ07. Oviposition was extremely affected by treatment with the NJ series compounds; at concentrations of 25 μM and 50 μM, oviposition reached almost zero with NJ05, NJ07 or NJ05 + NJ07 already at day one. Tested genes involved in egg biosynthesis were all confirmed by qPCR as downregulated in females treated with 25 μM NJ05 for 2 days, with a significant reduction in expression of p14, Tyrosinase 2, p48 and fs800. NJ05, NJ07 or NJ05 +NJ07 treatment of HEK293 (human embryonic cell line) and HES (human epithelial cell

**Funding:** This work was supported in by a grant from Fundação de Amparo à Pesquisa do Estado de São Paulo (FAPESP) to SVA. This study was financed in part by Coordenação de Aperfeiçoamento de Pessoal de Nível Superior (CAPES), Brazil – Finance Code 001. Fellowships from FAPESP have supported ASAP (2016/10046-6) and GOS (2018/24015-0). MSA was supported by Fundação Butantan. SMVA, JFO and MCAL were supported by fellowship PNPD from CAPES, process BCT-0149-4.03/17. SVA was also supported by institutional funds from Fundação Butantan. SVA is the recipient of an established investigator fellowship award from Conselho Nacional de Desenvolvimento Cientifico e Tecnologico, Brasil. The funders had no role in study design, data collection and analysis, decision to publish or preparation of the manuscript.

**Competing interests:** The authors have declared that no competing interests exist.

line) showed EC50 in the range of 18.42 to 145.20 µM. Overall, our results demonstrate that those molecules are suitable targets for further development into new drugs for schistosomiasis treatment, although progress is needed to lessen the cytotoxic effects on human cells. According to the present study, thiazole derivatives have schistosomicidal activities and may be part of a possible new arsenal of compounds against schistosomiasis.

## Introduction

Schistosomiasis is recognized as a serious public health problem in the world [1,2]. According to the World Health Organization (WHO), it is estimated that more than 206 million people worldwide are affected by the disease and that more than 700 million people live in areas where the disease is endemic [3]. Schistosomiasis is an infection caused by trematodes of the genus Schistosoma and occupies, after malaria, the second position in the world among parasitic infections [4,5]. This chronic disease is prevalent in tropical and subtropical regions, mainly in regions that lack basic sanitation, allowing the proliferation of contaminating agents, as well as the continuity of people being infected [2,3,6]. In addition, the absence of campaigns that promote health education to the population, as well as the lack of dissemination of epidemiological data, favor the continuity of the disease [6,7].

The current treatment for schistosomiasis is centered around praziquantel, an acylated quinoline-pyrazine derivative, that is able to act against the main strains of *Schistosoma mansoni* that infect humans and has been used for >30 years [8]. However, the exclusive use of praziquantel has been accompanied with reports in the literature that parasites isolated from some patients have been resistant to pharmacological treatment [9–11]. Besides, praziquantel has no effect on immature schistosomes and cannot prevent reinfection or alter the schistosome life cycle [12,13]. If it is considered that there is no effective alternative for the control of schistosomiasis, it is evident that the risk of developing resistance to praziquantel will increase in the coming decades, being a concern for the medical community. In view of this scenario, research priorities for facing schistosomiasis include the development and testing of new drugs, test combinations of praziquantel with new drugs as well as monitoring the spread of praziquantel drug resistance [1].

In view of the need to obtain alternatives for the treatment of schistosomiasis and other neglected diseases, efforts have been made within the medicinal chemistry, using various planning techniques to obtain substances with improved structural characteristics and with less toxicity [14,15]. These attempts comprehend praziquantel chemical structure modification by synthesis and evaluation of its analogues, rational design of new pharmacophores and discovery of new active compounds from screening programs on a large scale [16].

A class of derivatives that has gained attention because of its promising experimental antischistosomal activity is the one bearing the thiazole nucleus, which is effective against both immature and adult *S. mansoni* worms [17]. The thiazole moiety plays an important role in the medicinal chemistry field since it serves as a scaffold for new drug synthesis, being already used as antimicrobial (sulfathiazole), anthelmintic and fungicide (thiabenzadole) and schistosomicide (niridazole) compounds [18–20]. Niridazole (Ambilhar®) started being used for treatment in 1964, but patients had little tolerance to this drug with serious side effects due to the presence of nitroaromatic chemical groups that act as parasitophores (their presence is essential to kill the worm) and also cause damage to host DNA by formation of adducts that induce mutagenesis [16]. However, advances in the development of medicinal chemistry tools

have recently allowed the synthesis of new thiazole derivatives with anti-schistosomal activity. As an example, Santiago *et al*., 2014 [21] synthetized a set of molecules whose structures have a hydrazine and/or thiazole nucleus as a common group, and different pharmacophores including thiosemicarbazone, phthalyl thiosemicarbazone, phthalyl thiazole, and phthalyl thiazolidinone. Among them, the compounds comprised of thiazole and phthalimide were the ones that exhibited the highest activity against worms, with a significant decline in motility, pairing and oviposition, as well as a mortality rate of 100% after 144 h of treatment.

Recently, Reddy *et al*., 2018 [22] described the synthesis of a library of hydrazinyl-thiazoles via a three-component reaction of various aldehydes/ketones with thiosemicarbazide and different phenacyl bromides, but the authors did not evaluate any biological potential of the derivatives. Based on the potentially promising features of these new compounds, here we selected eight aryl-thiazole derivatives and examined their efficacy in terms of (i) schistosome survival, (ii) egg output (oviposition), (iii) motility, (iv) couples pairing, (v) ultrastructural alterations in the tegument of *S. mansoni* as determined by scanning electron microscopy (SEM), and (v) expression of *S. mansoni* developmental genes.

## Materials and methods

### Chemistry

Synthesis of the compounds bioassayed in this study was previously described [22–24]. Production of aryl-thiazole derivatives was carried out at the Laboratory of Chemistry and Therapeutic Innovation, at the Federal University of Pernambuco. Synthesis was carried out in two steps (Fig 1).

In the first step, the thiosemicarbazides (3.2 mmol) reacted in equimolar amounts with 2,4-dimethoxy-benzaldehyde (right side route) or 3,4,5-trimethoxy-benzaldehyde (left side route) through a condensation reaction in the presence of hydrochloric acid as the reaction catalyst to obtain a thiosemicarbazone (TSC), which served as an intermediary for thiazole synthesis following its acquisition. This second step was carried out in the presence of 2-(2,4-dimethoxybenzylidene)thiosemicarbazone (0.62 mmol) or 2-(3,4,5-trimethoxybenzylidene)thiosemicarbazone (0.55 mmol), 1.1 molar equivalent of different 2-chloro or 2-bromoacetophenones (unsubstituted, 4-chloro-phenyl, 4-bromo-phenyl, 4-nitro-phenyl) under basic conditions (4.0 molar equivalent of sodium acetate) and ethanol in reflux [25].

Reactions were monitored with analytical thin-layer chromatography in silica gel 60 F254 plates and visualized under UV light (254 nm). Melting points were determined on a Quimis 340 capillary melting point apparatus and were not corrected. IR spectra were measured on Bruker IFS-66 IR spectrophotometer (Bruker, Germany) using KBr pellets. NMR spectra were recorded on Bruker Avance (600 MHz for $^1$H and 150 MHz for $^{13}$C) instruments. HRMS were performed on a MALDI-TOF Autoflex III (Bruker Daltonics, Billerica, MA, USA). The spectroscopic data of the synthesized compounds (NJ03 to NJ08, NJ11 and NJ12) are in accordance with the structures proposed in Fig 1 and with the spectra published by Reddy *et al*., *2018* [22].

**NJ03: (E)-4-(4-chlorophenyl)-2-((E)-(2,4-dimethoxybenzylidene)hydrazono)-2,3-dihydrothiazole.** $C_{18}H_{16}ClN_3O_2S$. Yield: 71.20%. M.P.: 191-192°C. Rf: 0.44 (7:3 *n*-hexane/ethyl acetate). NMR $^1$H (600 MHz, DMSO) δ: 3.80 (s, 6H, $OCH_3$); 6.60 (m, 2H, CH-Ar); 7.35 (s, 1H, thiazole); 7.46 (d, 2H, CH-Ar, J = 12 Hz); 7.71 (d, 1H, CH-Ar, J = 6 Hz); 7.83 (d, 2H, CH-Ar, J = 12 Hz); 8.38 (s, 1H, HC = N); 12.30 (s, 1H, NH). NMR $^{13}$C (DMSO, 150 MHz) δ: 55.94, 56.28, 104.85, 107.07, 115.38, 126.76, 127.93, 129.10, 129.99, 132.86, 139.66, 147.45, 159.19, 162.68, 168.94. IR (KBr, cm$^{-1}$): 1396 (N = C-S), 1459 (C = N), 3317 (NH). HRMS *m/z* [*M*+H]$^+$ calcd for $C_{18}H_{16}ClN_3O_2S$: 374.069; found: 374.079.

**Fig 1. Route of synthesis to obtain the aryl-thiazole derivatives (NJ series).** In the compounds, R was hydrogen, 4-chloro, 4-bromo or 4-nitro. For 2,4-dimethoxy-aryl thiazole (right side route) R = Cl for NJ03; R = NO$_2$ for NJ06; R = Br for NJ08 and R = H for NJ12. For 3,4,5-trimethoxy-aryl thiazole (left side route) R = Cl for NJ04; R = NO$_2$ for NJ05; R = Br for NJ07 and R = H for NJ11.

**NJ04: (E)-4-(4-chlorophenyl)-2-((E)-(3,4,5-trimethoxybenzylidene)hydrazono)-2,3-dihydrothiazole.** C$_{19}$H$_{18}$ClN$_3$O$_3$S. Yield: 61.80%. M.P.: 135-136°C. Rf: 0.56 (7:3 *n*-hexane/ethyl acetate). NMR $^1$H (600 MHz, DMSO) δ: 3.70 (s, 3H, OCH$_3$); 3.87 (s, 6H, OCH$_3$); 6.99 (s, 2H, CH-Ar); 7.38 (s, 1H, thiazole); 7.47 (d, 2H, CH-Ar, J = 6 Hz); 7.88 (d, 2H, CH-Ar, J = 6 Hz); 7.97 (s, 1H, HC = N); 12.05 (s, 1H, NH). NMR $^{13}$C (DMSO, 150 MHz) δ: 56.34, 60.59, 104.05, 104.86, 127.67, 129.08, 130.40, 132.39, 134.06, 139.18, 141.76, 149.83, 153.67, 168.87. IR (KBr, cm$^{-1}$): 1326 (N = C-S), 1411 (C = N), 3294 (NH). HRMS *m/z* [*M*+H]$^+$ calcd for C$_{19}$H$_{18}$ClN$_3$O$_3$S: 404.079; found: 404.081.

**NJ05: (E)-4-(4-nitrophenyl)-2-((E)-(3,4,5-trimethoxybenzylidene)hydrazono)-2,3-dihydro-thiazole.** C$_{19}$H$_{18}$N$_4$O$_5$S. Yield: 55.20%. M.P.: 212-214°C. Rf: 0.62 (6:4 *n*-hexane/ethyl acetate). NMR $^1$H (600 MHz, DMSO) δ: 3.69 (s, 3H, OCH$_3$); 3.86 (s, 6H, OCH$_3$); 6.97 (s, 2H, CH-Ar); 7.71 (s, 1H, thiazole); 8.00 (s, 1H, HC = N); 8.10 (d, 2H, CH-Ar, J = 6 Hz); 8.26 (d, 2H, CH-Ar, J = 6 Hz); 12.34 (s, 1H, NH). NMR $^{13}$C (DMSO, 150 MHz) δ: 56.35, 60.59, 104.12, 109.00, 124.54, 126.79, 129.18, 130.28, 139.25, 141.07, 142.22, 146.66, 153.66, 165.09, 169.13. IR (KBr, cm$^{-1}$): 1345 (N = C-S), 1412 (C = N), 3305 (NH). HRMS *m/z* [*M*+H]$^+$ calcd for C$_{19}$H$_{18}$N$_4$O$_5$S: 415.103; found: 415.109.

**NJ06: (E)-2-((E)-(2,4-dimethoxybenzylidene)hydrazono)-4-(4-nitrophenyl)-2,3-dihy-drothiazole.** C$_{18}$H$_{16}$N$_4$O$_4$S. Yield: 72.60%. M.P.: 218-220°C. Rf: 0.68 (7:3 *n*-hexane/ethyl acetate). NMR $^1$H (600 MHz, DMSO) δ: 3.81 (s, 6H, OCH$_3$); 6.61 (m, 2H, CH-Ar); 7.64 (s, 1H, thiazole); 7.70 (d, 1H, CH-Ar, J = 12 Hz); 8.08 (d, 2H, CH-Ar, J = 6 Hz); 8.24 (d, 2H, CH-Ar, J = 12 Hz); 8.28 (s, 1H, HC = N); 12.09 (s, 1H, NH). NMR $^{13}$C (DMSO, 150 MHz) δ: 55.86, 56.20, 106.95, 108.56, 115.63, 124.49, 126.54, 126.76, 138.15, 141.16, 146.61, 148.88, 158.96, 162.40, 169.23. IR (KBr, cm$^{-1}$): 1325 (N = C-S), 1413 (C = N), 3172 (NH). HRMS *m/z* [*M*+H]$^+$ calcd for C$_{18}$H$_{16}$N$_4$O$_4$S: 385.093; found: 386.100.

**NJ07: (E)-4-(4-bromophenyl)-2-((E)-(3,4,5-trimethoxybenzylidene)hydrazono)-2,3-dihydrothiazole.** $C_{19}H_{18}BrN_3O_3S$. Yield: 65.80%. M.P.: 159-161ºC. Rf: 0.45 (6:4 *n*-hexane/ethyl acetate). NMR $^1$H (600 MHz, DMSO) δ: 3.70 (s, 3H, OCH$_3$); 3.87 (s, 6H, OCH$_3$); 6.98 (s, 2H, CH-Ar); 7.39 (s, 1H, thiazole); 7.61 (d, 2H, CH-Ar, J = 6 Hz); 7.81 (d, 2H, CH-Ar, J = 6 Hz); 7.97 (s, 1H, HC = N); 12.22 (s, 1H, NH). NMR $^{13}$C (DMSO, 150 MHz) δ: 56.33, 56.52, 60.59, 104.05, 104.96, 120.97, 127.99, 130.39, 131.99, 134.38, 139.17, 141.79, 149.85, 153.66, 168.88. IR (KBr, cm$^{-1}$): 1347 (N = C-S), 1409 (C = N), 3305 (NH). HRMS *m/z* [*M*+H]$^+$ calcd for $C_{19}H_{18}BrN_3O_3S$: 448.029; found: 448.032.

**NJ08: (E)-4-(4-bromophenyl)-2-((E)-(2,4-dimethoxybenzylidene)hydrazono)-2,3-dihydrothiazole.** $C_{18}H_{16}BrN_3O_2S$. Yield: 37.10%. M.P.: 173-174ºC. Rf: 0.57 (7:3 *n*-hexane/ethyl acetate). NMR $^1$H (600 MHz, DMSO) δ: 3.80 (s, 6H, OCH$_3$); 6.60 (m, 2H, CH-Ar); 7.36 (s, 1H, thiazole); 7.60 (d, 2H, CH-Ar, J = 6 Hz); 7.71 (d, 1H, CH-Ar, J = 12 Hz); 7.77 (d, 2H, CH-Ar, J = 12 Hz); 8.36 (s, 1H, HC = N); 12.20 (s, 1H, NH). NMR $^{13}$C (DMSO, 150 MHz) δ: 55.93, 56.27, 104.87, 107.05, 115.36, 126.71, 128.18, 130.26, 131.73, 132.00, 146.90, 159.13, 162.60, 167.61, 168.94. IR (KBr, cm$^{-1}$): 1278 (N = C-S), 1421 (C = N), 3312 (NH). HRMS *m/z* [*M*+H]$^+$ calcd for $C_{18}H_{16}BrN_3O_2S$: 418.018; found: 418.030.

**NJ11: (E)-4-phenyl-2-((E)-(3,4,5-trimethoxybenzylidene)hydrazono)-2,3-dihydrothiazole.** $C_{19}H_{19}N_3O_3S$. Yield: 54.00%. M.P.: 193-195ºC. Rf: 0.52 (7:3 *n*-hexane/ethyl acetate). NMR $^1$H (600 MHz, DMSO) δ: 3.70 (s, 3H, OCH$_3$); 3.75 (s, 6H, OCH$_3$); 6.99 (s, 2H, CH-Ar); 7.31 (d, 2H, CH-Ar, J = 12 Hz); 7.40 (s, 1H, thiazole); 7.42 (d, 1H, CH-Ar, J = 12 Hz); 7.87 (d, 2H, CH-Ar, J = 6 Hz);7.98 (s, 1H, HC = N); 12.22 (s, 1H, NH). NMR $^{13}$C (DMSO, 150 MHz) δ: 56.22, 56.34, 60.60, 104.03, 125.98, 127.99, 129.07, 130.46, 135.17, 139.14, 141.61, 150.98, 153.67, 168.72. IR (KBr, cm$^{-1}$): 1324 (N = C-S), 1413 (C = N), 3176 (NH). HRMS *m/z* [*M*+H]$^+$ calcd for $C_{19}H_{19}N_3O_3S$: 370.118; found: 370.125.

**NJ12: (E)-4-phenyl-2-((E)-(3,4,5-trimethoxybenzylidene)hydrazono)-2,3-dihydrothiazole.** $C_{18}H_{17}N_3O_2S$. Yield: 61.40%. M.P.: 220-222ºC. Rf: 0.46 (7:3 *n*-hexane/ethyl acetate). NMR $^1$H (600 MHz, DMSO) δ: 3.81 (s, 6H, OCH$_3$); 6.62 (m, 2H, CH-Ar); 7.27 (s, 1H, thiazole); 7.29 (d, 1H, CH-Ar, J = 12 Hz); 7.40 (t, 2H, CH-Ar, J = 6 Hz); 7.73 (d, 1H, CH-Ar, J = 6 Hz); 7.86 (d, 2H, CH-Ar, J = 6 Hz); 8.28 (s, 1H, HC = N); 11.98 (s, 1H, NH). NMR $^{13}$C (DMSO, 150 MHz) δ: 55.88, 56.21, 103.65, 106.97, 115.85, 126.00, 126.51, 128.45, 129.03, 135.28, 137.56, 151.01, 158.89, 162.29, 168.87. IR (KBr, cm$^{-1}$): 1331 (N = C-S), 1417 (C = N), 3305 (NH). HRMS *m/z* [*M*+H]$^+$ calcd for $C_{18}H_{17}N_3O_2S$: 340.108; found: 340.115.

## Ethics statement

The experimental protocols were in accordance with the Ethical Principles in Animal Research adopted by the Brazilian College of Animal Experimentation (COBEA) and the protocol/experiments have been approved by the Ethics Committee for Animal Experimentation of Institute Butantan (CEUA Nº 1777050816).

## Maintenance of parasite life cycle

The BH strain of *S. mansoni* (Belo Horizonte, Brazil) was maintained in the intermediate snail host *Biomphalaria glabrata* and the golden hamster (*Mesocricetus auratus*) was used as definitive host. Female hamsters aged 3–4 weeks, freshly weaned, weighing 50–60 g, were housed in cages (30x20x13cm) containing a sterile bed of wood shavings. A standard diet (Nuvilab CR-1 Irradiada, Quimtia S/A, Paraná, Brazil) and water were made available *ad libitum*. The room temperature was kept at 22 ± 2°C and a 12:12 hour light–dark cycle was maintained. Hamsters were infected by exposure to a *S. mansoni* cercariae suspension containing approximately 200–250 cercariae using the ring technique [26]. After 49 days of infection, the *S. mansoni*

adult worms were recovered by perfusion of the hepatic portal system [27]. Cercariae were released from infected snails and mechanically transformed to obtain schistosomula *in vitro* [28].

## Treatment of schistosomula and adult worms with NJ series compounds

Newly transformed schistosomula (NTS) were maintained for 2 h in M169 (Vitrocell) medium supplemented with 2% fetal bovine serum (FBS) (Vitrocell), 1 μM serotonin, 0.5 μM hypoxanthine,1 μM hydrocortisone, 0.2 μM triiodothyronine, penicillin/streptomycin, amphotericin, gentamicin (Vitrocell) at 37˚C and 5% $CO_2$ [28]. Only after 2 h incubation in culture medium the NJ series treatment was initiated as follows: 6.25 μM, 12.5 μM, 25 μM or 50 μM compound, for 1–5 days. Paired adult worms freshly perfused from infected hamsters (see above) were maintained in culture in RPMI medium (Gibco) supplemented with 10% fetal bovine serum (FBS) (Vitrocell), penicillin/streptomycin, amphotericin, gentamicin (Vitrocell) at 37˚C and 5% $CO_2$ for 2 h prior to the beginning of treatment with NJ series compound as follows: 12.5 μM, 25 μM or 50 μM compound, for 1–3 days of treatment. In all cases, NJ series compounds were prepared from a stock solution of 20 mM in dimethyl sulfoxide (DMSO), and the equivalent amount of DMSO was added to the control assays.

## Motility assay of adult worms

To evaluate the general condition of adult worms, including motility and mortality rate an inverted microscope was used. Parasites were observed after 1, 2 and 3 days of treatment with different concentrations of NJ series compounds or vehicle DMSO at 0.1%. Motility and survival of worms were assessed according to the criteria scored in a viability scale of 0–3 [29]. The scoring system was as follows: 3, complete body movement; 1.5, partial body movement or immobile but alive; and 0, dead. Treatment was considered lethal whenever no worm movement was detected when observed for 2 min. Oviposition and pairing status were also observed and eggs were counted.

## Viability assay of schistosomula and adult worms

Viability of schistosomula and *S. mansoni* adult worms after treatment was determined by a cytotoxicity assay based on the CellTiter-Glo® Luminescent Cell Viability Assay kit (G7570, Promega, Madison, Wisconsin, EUA) [30]. The assay determines the amount of ATP present in freshly lysed adults or in intact schistosomula; the assay signals the presence of metabolically active cells.

In addition, the viability of schistosomula was evaluated by the presence or absence of dead parasites, through staining with propidium iodide (PI) [31]. Schistosomula were equally distributed in 96-well microtiter plates, incubated with different concentrations of NJ05 + NJ07 compounds indicated in the figure or the corresponding DMSO vehicle (control), and 2 μg/mL propidium iodide (PI) (Sigma-Aldrich) were added at 2 days of drug exposure. The parasites were immediately observed with light microscopy at 10 x magnification using a Nikon Eclipse fluorescence inverted microscope. Under fluorescence microscopy, schistosomula death was scored by a red fluorescence signal (572 nm emission microscope filter) [31,32]. The number of biological replicates that were assayed, as well as the number of parasites that were counted per replicate, is stated in the legends to the figures.

## Cytotoxicity evaluation of NJ05, NJ07 or NJ05 + NJ07 in HEK293 and HES human cell lines

HEK293 (Human Embryo Kidney- HEK-293 -ATCC CRL-1573) and HES (human endometrial epithelial cell line, kindly provided by Prof. Douglas Anthony Kniss—Ohio State University, USA) [33] cell cultures ($5x10^3$ cells/well) were incubated for two days with different concentrations of NJ05, NJ07 or NJ05+NJ07 (3.125 μM—400 μM), and after treatment their viability was measured by ATP quantitation with the CellTiter-Glo Luminescent Cell Viability Assay (G7570, Promega, Madison, Wisconsin, EUA). The data was used for calculating the EC50 with Origin software (OriginLab, Northampton, MA). All experiments were performed with 3 biological replicates.

## Scanning electron microscopy (SEM)

Adult worms collected by perfusion were immediately transferred to supplemented RPMI medium; parasites were distributed in 6-well plates (adults: 10 paired worm couples per well) with medium. The worms were kept in culture (oven at 37˚C and 5% $CO_2$) for 2 h for adaptation, and then NJ series compounds or the equivalent amount of 0.1% DMSO vehicle were added.

Ultrastructural analysis was performed with scanning electron microscopy. Adult worms incubated at different concentrations of NJ series compounds or with 0.1% DMSO vehicle for 1 and 2 days were fixed with modified Karnovsky reagent (1% paraformaldehyde, 2.5% glutaraldehyde, 1 mM calcium chloride in 0.1 M sodium cacodylate buffer, pH 7.4) and after the fixing stage the material was washed with sodium cacodylate buffer (0.1 mol / L, pH 7.2) and post-fixed with 1% osmium tetroxide (w / v) for 1 h.

Samples were dehydrated with increasing concentrations of ethanol and then dried with liquid $CO_2$ in a critical-point dryer machine (model Leica EM CPD030, Leica Microsystems, Illinois, USA). Treated specimens were mounted on aluminum microscopy stubs and coated with gold particles using an ion-sputtering apparatus (model Leica EM SCD050, Leica Microsystems, Illinois, USA) [34]. Specimens were then observed and photographed using an electron microscope (FEI QUANTA 250, *Thermo Fisher Scientific*, *Oregon*, USA).

## RT-qPCR validation of *S. mansoni* developmental genes

For quantitative RT-PCR, complementary DNAs were obtained by reverse transcription (RT) of 50 ng adult worms total RNA using SuperScript IV Reverse Transcriptase (Invitrogen) and random hexamer primers in a 20 μL volume, according to the manufacturer's recommendations. The resulting cDNA was diluted 8-fold in water and qPCR amplification was done with 2.5 μL of diluted cDNA in a total volume of 10 μL using SYBR Green Master Mix (Life Technologies) and specific primer pairs (S1 Table). Tested genes were described in previous publications from the literature [35–40] or selected here based on their predicted function; primers were designed with the use of Primer3 online software. The Light cycle 480 II (Roche) qPCR was used. Results were analyzed by comparative Ct method and the statistical significance was calculated with the two-sided t-test.

To find adequate normalizer genes for qPCR, we looked for evidence of genes with non-detectable changes in expression upon NJ series compounds treatment; ten genes were selected based on literature reports [35–40] and tested by qPCR in the adult female worms for the lack of effect of NJ series compounds on their levels of expression. The two least affected genes were Mitochondrial 28s ribosomal protein s14 (Smp_090920) and PI3K regulatory subunit 4

(Smp_123610). In all cases, the geometric mean expression of these two genes was used as normalizer for calculating the expression levels of genes of interest.

## Results

### Phenotypic effects of NJ series compounds on *S. mansoni* schistosomula

Eight different dimethoxy-aryl thiazole derivatives (see Fig 1) were tested for schistosomicidal properties against schistosomula early forms of the parasite. For this, schistosomula were first mechanically transformed from cercariae and were pre-incubated for 2 h in culture medium prior to compound addition. After each compound was added, schistosomula viability was evaluated every day for a period of five days by measuring the concentration of ATP in the metabolically active cells of the parasite. Different concentrations of each compound from 6.25 to 50 µM were tested, and out of the eight compounds that were assayed (Fig 2A and Fig A in S1 Text), two had a more prominent effect in reducing schistosomula viability, namely NJ05 and NJ07 (Fig 2A). These compounds decreased the parasites' viability by approximately 40% after five days of exposure (Fig 2A). In order to test for a combinatorial effect, the best two compounds were assayed together. Within the first day of treatment with NJ05 + NJ07, across all concentrations tested (12.5, 25 or 50 µM) there was a dose-dependent decrease of 60 to 80% in parasite viability (Fig 2B), indicating a combinatorial effect between the two compounds. The phenotype was more prominent after five days of treatment, with viability reduced to zero (Fig 2B).

The effect on schistosomula viability over 5 days of exposure to each of the other six NJ series compounds is shown at Fig A in S1 Text; a less marked 10 to 40% reduction in parasite viability was observed.

Propidium iodide (PI) was used as an alternate method to quantitate viability of schistosomula after a 2-days incubation in the presence of control vehicle (0.1% DMSO) or of 6.25 to 50 µM of the best two compounds together (NJ05 + NJ07) (Fig 2C and 2D). Under fluorescence microscopy, dead schistosomula were detected by a red fluorescence signal from PI (536 nm emission microscope filter) (Fig 2D, lower panels). Total number of schistosomula present in the field was detected and counted under light optical microscopy at each NJ05 + NJ07 concentration (Fig 2D, upper panels), and the percentage viability was computed and plotted as shown in Fig 2C; mortality was 100% with 25 or 50 µM NJ05 + NJ07. Incubation with control vehicle (0.1% DMSO) for 2 days showed no reduction in parasite viability (Fig 2B and 2D, first upper and lower panels).

### Effects of NJ series compounds on *S. mansoni* adult worms' viability and motility

In order to evaluate the schistosomicidal activity of all eight compounds on adult worms the oviposition, male-female pairing status, worms' viability, motility, and alterations in the tegument were measured. Compounds were tested at a concentration range of 12.5 to 50 µM and observed every day for a period of 3 days.

Again, compounds NJ05, NJ07 or NJ05 + NJ07 exhibited the highest schistosomicidal activities among the eight tested, showing a concentration-dependent significant decrease in viability of the adult worms, as evaluated by ATP quantitation (Fig 3A–3C). Compound NJ05 was more effective than NJ07, and viability after three days was reduced to 25% in parasites treated with 50 µM NJ05 (Fig 3A), compared with a viability reduction to 60% with 50 µM NJ07 (Fig 3B). No combinatorial effect was detected when the two compounds were tested together (NJ05 + NJ07), as the viability was reduced to approximately 25% after three days of exposure

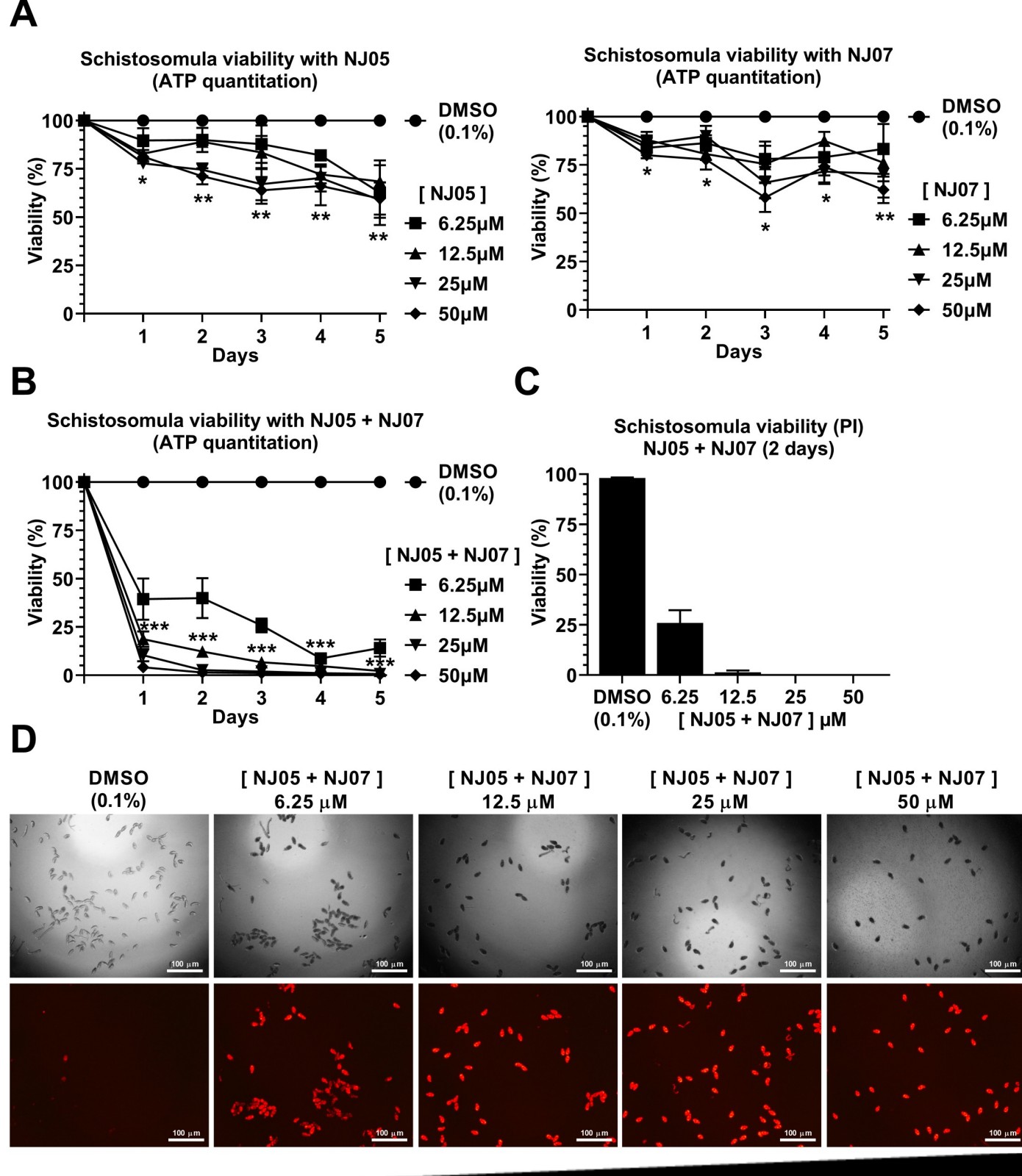

**Fig 2. Effect of NJ05, NJ07 or NJ05 + NJ07 on schistosomula viability. (A and B)** ATP quantitation using a luminescent assay to assess schistosomula viability under NJ series compounds exposure. Schistosomula (100-120/well) were incubated with the indicated concentrations of NJ05, NJ07, NJ05 + NJ07 or with vehicle

(0.1% DMSO) for up to 5 days. Viability was expressed as % luminescence values relative to the control (DMSO). Mean ± SEM from three biological replicate experiments are shown. The two-way ANOVA test was used to calculate the statistical significance ($^*p < 0.05$; $^{**}p < 0.01$; $^{***}p < 0.001$). For clarity purposes, we show only the highest p-value obtained from the two-way ANOVA test for each time point on all different concentrations tested. **(C)** Quantitation of schistosomula viability using propidium iodide staining; schistosomula were treated for two days with NJ05 + NJ07 at the different concentrations indicated. Percentage of viable schistosomula (non-stained with propidium iodide) is shown. For each condition tested, about 400 schistosomula were used, divided into four biological replicates. Mean ± SEM from four replicate experiments are shown. **(D)** Schistosomula treated with the indicated concentrations of NJ05 + NJ07 or with vehicle (0.1% DMSO) for 2 days were visualized by staining with propidium iodide (marker of dead cells; 572 nm emission filter microscope). For each concentration (indicated at the top), the upper panel shows a light microscopy image and the bottom panel shows the image of the same field with differential fluorescence detection of PI-positive parasites. Bar = 100 μm.

to 50 μM NJ05 + NJ07 (Fig 3C), similar to the reduction caused by NJ05 only. The effect of other tested compounds is shown at Fig B in S1 Text; after three days of treatment with compounds NJ03, NJ04 or NJ08 (at 50 μM concentration), a reduction in viability to only 75% was observed.

*S. mansoni* adult worms exhibited a larger decrease in motility when exposed to NJ05 or NJ05 + NJ07, compared with NJ07 treatment alone. This phenotype was dependent upon concentration and incubation time (Fig 3D–3F). After three days with the lowest concentration tested (12.5 μM), NJ05 or NJ05 + NJ07 decreased the motility to 40–50% compared with controls (Fig 3D and 3F), while NJ07 treatment at the same concentration led to a reduction of only 15–20% in motility (Fig 3E). Impairment of the peristaltic movement and reduced ability of the suckers to adhere to the bottom of the culture plates were observed. Among the other compounds of the NJ series tested we can highlight the 50% decrease in motility of adult worms exposed to 50 μM NJ03 or NJ08 after three days of exposure (Fig C in S1 Text).

## Effects of NJ series compounds on *S. mansoni* adult worm couples pairing and female oviposition

Compound NJ05 showed a more pronounced effect than NJ07 on worm couples pairing (Fig 4A and 4B). Thus, couples exposed for one day to 25 μM or 50 μM NJ05 already exhibited no pairing (Fig 4A), whereas there was no couple pairing reduction at day one with 25 μM NJ07, and with 50 μM NJ07, pairing was reduced to approximately 60% compared with control (Fig 4B). Combination of NJ05 + NJ07 was not effective in enhancing couples unpairing (Fig 4C) when compared with NJ05 alone. Among other NJ series compounds (see Fig D in S1 Text), it can be highlighted the unpairing of approximately 75% - 90% of adult worm couples after three days exposure to 50 μM NJ03, NJ04, NJ08 or NJ11.

In addition, oviposition was extremely affected by treatment with the NJ series compounds, especially NJ05, NJ07 or NJ05 + NJ07 (Fig 4D–4F). Of note, reduction in oviposition was independent from couple unpairing. Thus, at concentrations of 25 μM or 50 μM, oviposition reached almost zero with NJ05, NJ07 or NJ05 + NJ07 already at day one (Fig 4D–4F), even when unpairing was incomplete or not present such as with NJ07 (compare Fig 4B and 4E). Results for oviposition reduction of the other tested compounds are described at Fig E in S1 Text; little effect was observed in oviposition with these other compounds.

## EC50 assay of NJ05, NJ07 or NJ05 + NJ07 in HEK293 and HES human cell lines

Toxicity of the compounds was evaluated by ATP quantitation on human HEK293 embryonic kidney cell line and human HES endometrial epithelial cell line in culture, after exposure for two days to 3.125–200 μM NJ05, NJ07 or NJ05 + NJ07. The concentrations of NJ05, NJ07 or NJ05 + NJ07 that caused a 50% reduction in viability (EC50) of HEK293 cells were 46.64 μM, 92.35 μM, and 18.42 μM, respectively (Fig F in S1 Text). And for HES cell line the EC50 with

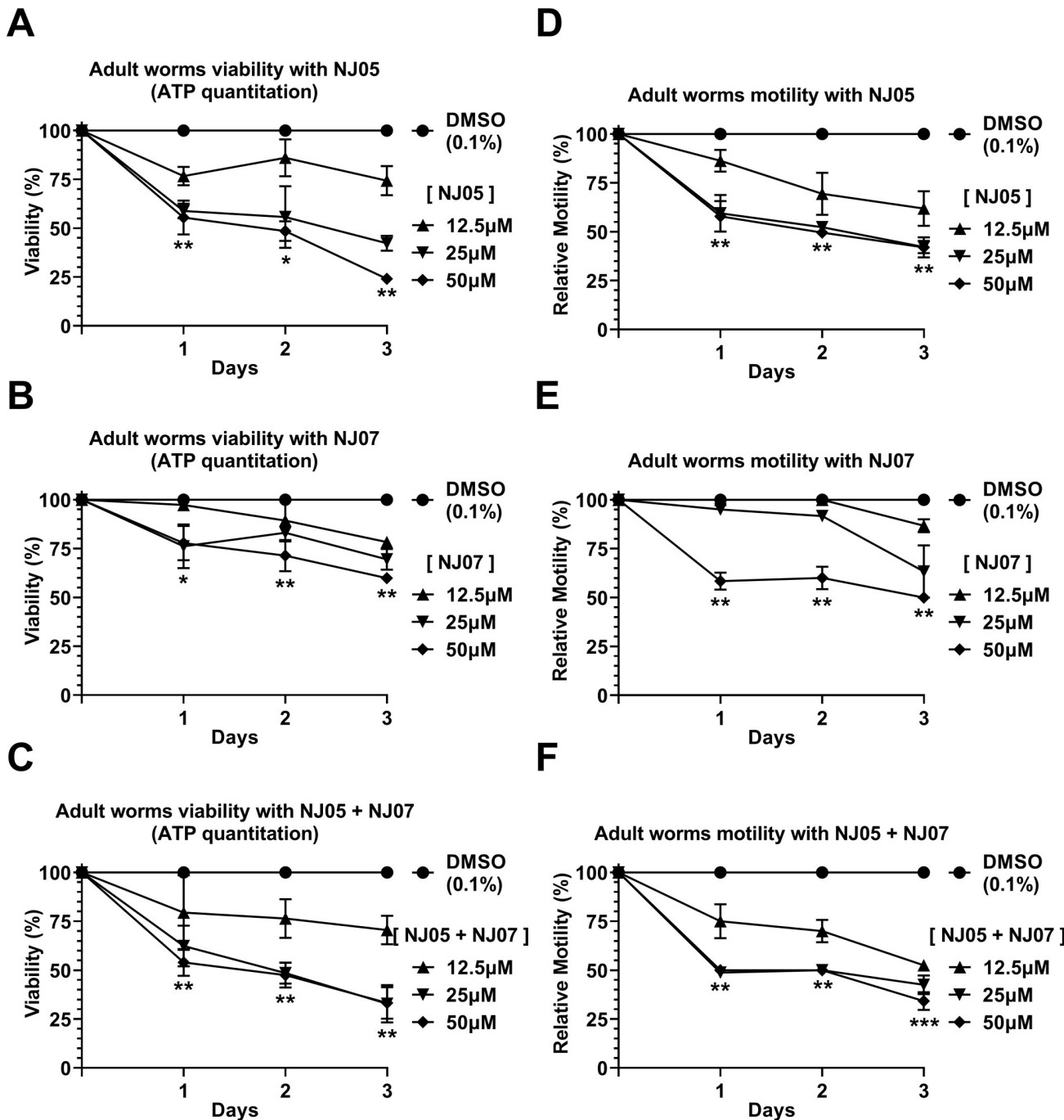

**Fig 3.** *In vitro* **effects of NJ05, NJ07 or NJ05 + NJ07 on the viability and motility of *Schistosoma mansoni* adult worms. (A—C)** Viability was estimated by the total amount of ATP available in the parasites, using a luminescent assay. Pairs of adult worms were treated for 2 days with NJ05, NJ07 or NJ05 + NJ07 at the different concentrations indicated or with vehicle (0.1% DMSO). Viability was expressed as percentage luminescence values relative to the control (DMSO). Mean ± SEM from three replicate experiments are shown, each with 10 worm pairs. **(D—F)** Percentage of relative motility of adult worms treated with different concentrations of NJ05, NJ07 or NJ05 + NJ07 and controls (0.1% DMSO) at different times of exposure from 1 to 3 days. Mean ± SEM of three experiments, each with 10 worm pairs. $^*p < 0.05$, $^{**}p < 0.01$ and $^{***}$ $p < 0.001$ compared with controls. For clarity purposes, we show only the highest p-value obtained from the two-way ANOVA test for each time point on all different concentrations tested.

NJ05, NJ07 or NJ05 + NJ07 were 133.53 μM, 145.20 μM and 29.59 μM, respectively (Fig F in S1 Text).

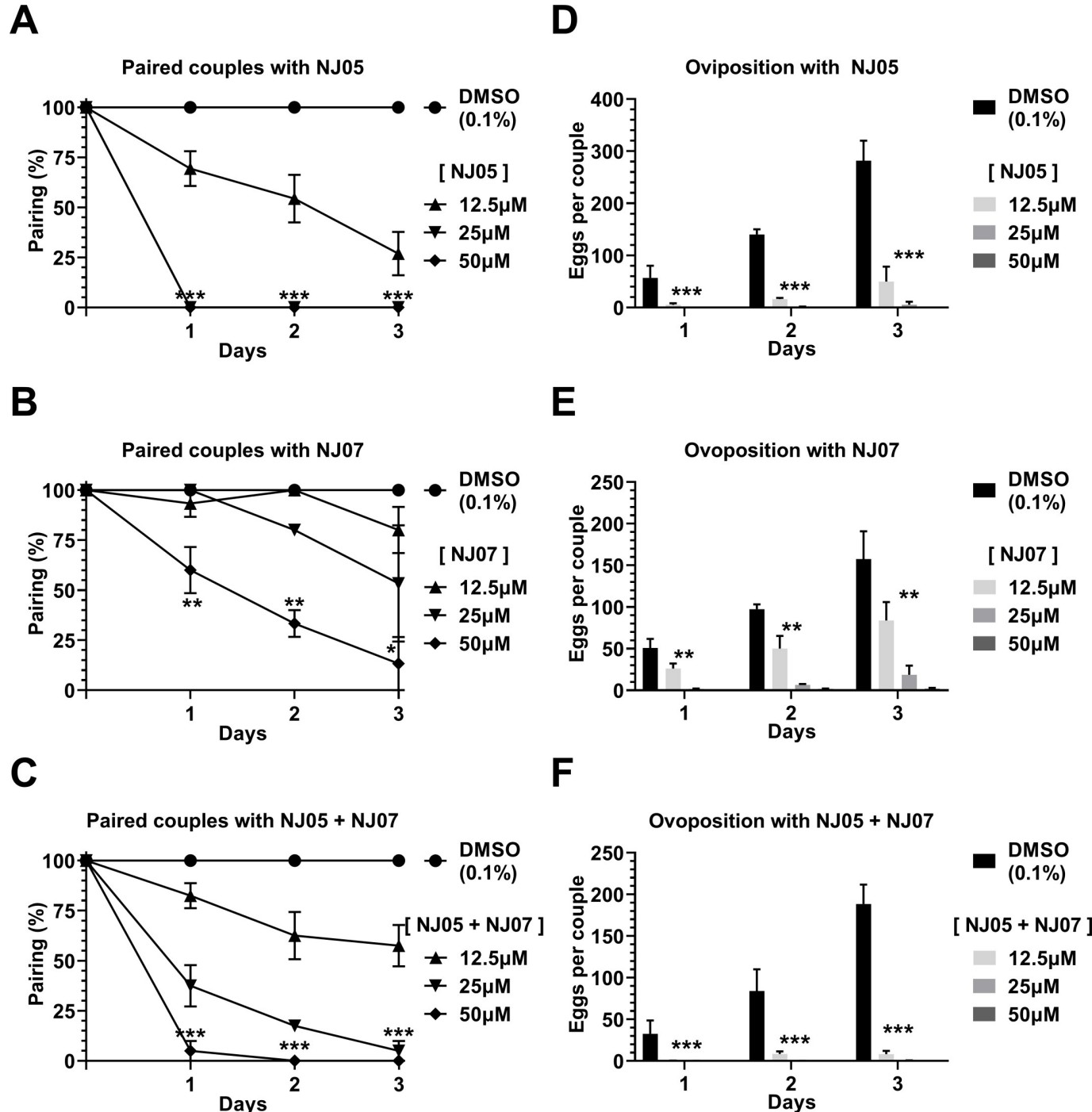

**Fig 4.** *In vitro* **effects of NJ05, NJ07 or NJ05 + NJ07 on pairing and oviposition of** *Schistosoma mansoni* **adult worms. (A-C)** We monitored the pairing of control couples and of couples treated with different concentrations of NJ05, NJ07 or NJ05 + NJ07 at different times of exposure from 1 to 3 days. Mean ± SEM of three experiments are shown, each with 10 worm pairs. **(D-F)** Number of eggs released by females incubated with NJ05, NJ07 or NJ05 + NJ07 at different concentrations or with vehicle (0.1% DMSO) at different times of exposure from 1 to 3 days. *p < 0.05, **p < 0.01 and *** p < 0.001. For clarity purposes, we show only the highest p-value obtained from the two-way ANOVA test for each time point on all different concentrations tested.

## Scanning electron microscopy of treated adult worms

Through scanning electron microscopy (SEM) it was possible to detect phenotypic alterations between control and treated adult worms. Control adult male schistosomes that were not exposed to any drugs are depicted in Fig 5 (panels 1–4) and Fig 6 (panels 1–4) and showed a normal topography of the surface membrane of anterior and dorsal regions, respectively. Oral and ventral suckers could be clearly visualized (Fig 5, panels 2–4). In the anterior portion of the body, the gynecophoral canal, a longitudinal fold of the middle and posterior body that houses the female for the purpose of mating and reproduction, could be identified (Fig 5, panel 2). In Fig 6, panels 1 to 4, the medial and posterior parts of schistosomes are shown, respectively. In these regions, a large number of tubercles with typical spines and ciliated papillae was observed (Fig 6, panels 2 and 3).

SEM revealed detailed surface membrane ultrastructural damage on the anterior (Fig 5) and dorsal (Fig 6) regions of adult male worms caused by *in vitro* exposure for 2 days to 25 µM NJ05 (Figs 5 and 6, panels 5–8) or NJ07 (Figs 5 and 6, panels 9–12) compared with controls (0.1% DMSO) (Figs 5, and 6, panels 1–4). Thus, incubation of adult male *S. mansoni* worms with NJ05 resulted in severe tegument destruction of the anterior region (Fig 5, panels 5–8). In

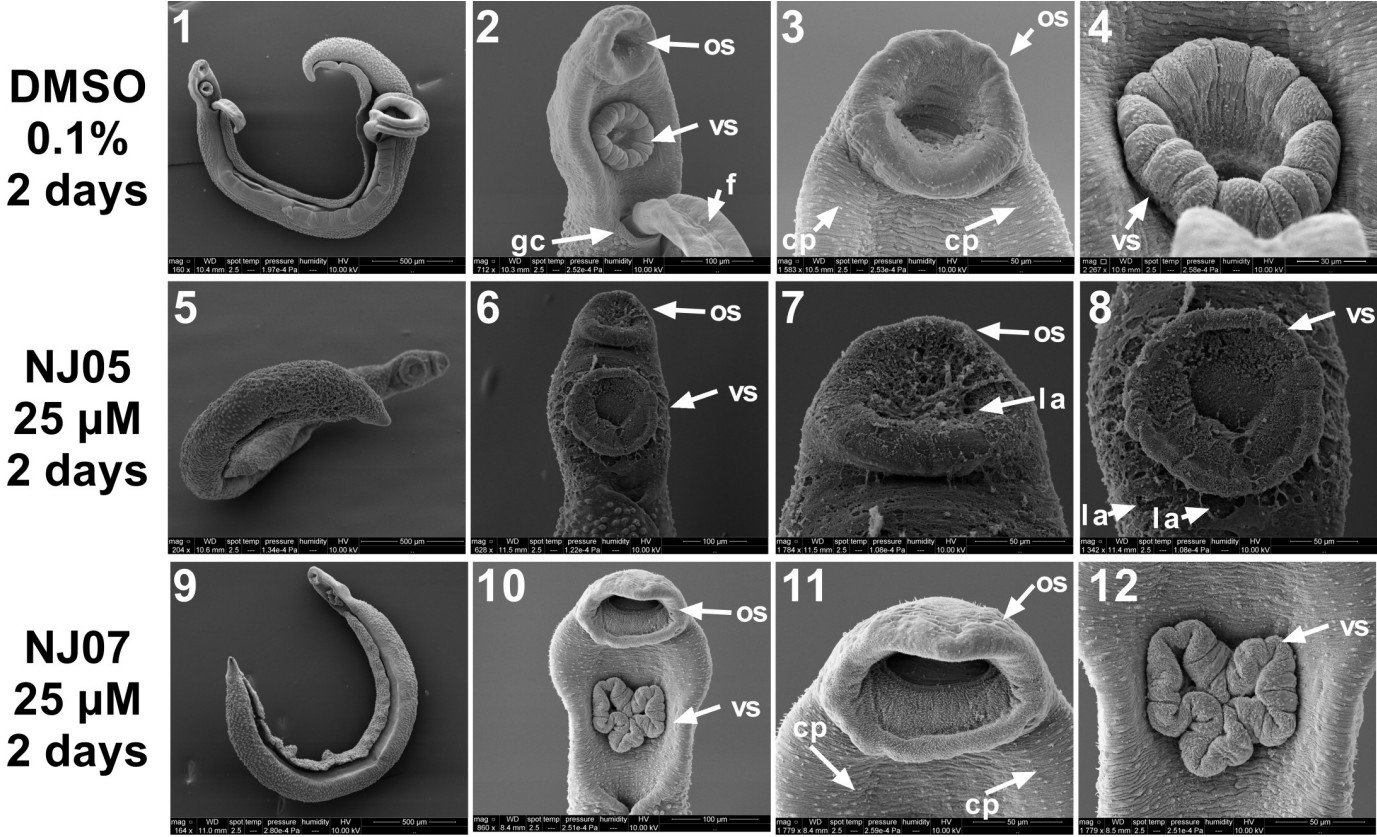

**Fig 5. Micrographs showing alterations on the ultrastructure of the anterior region of *Schistosoma mansoni* adult male worms exposed to NJ05 or NJ07.** Scanning electron microscopy of worms exposed for 2 days to 25 µM NJ05 (panels 5–8), 25 µM NJ07 (panels 9–12) or vehicle (0.1% DMSO) (panels 1–4). Note that the size scale bar is shown within the black thin line below each image; **panels 1 and 2**: Control worms (Bar = 500 µm; 100 µm) presented the fully paired couple; **panels 3 and 4**: Anterior region of control male presented normal oral and ventral suckers tegument (Bar = 50 µm; 30 µm); **panels 5 and 6**: Anterior region of adult male worm treated with 25 µM NJ05 (Bar = 500 µm; 100 µm); **panels 7 and 8**: Male oral and ventral suckers presented tegument peeling (Bar = 50 µm); **panels 9 and 10**: Anterior region of male adult worm treated with 25 µM NJ07 (Bar = 500 µm; 100 µm); **panels 11 and 12**: Male oral and ventral suckers presented tegument without structural alterations (Bar = 50 µm). *f*: female; *os*: oral sucker; *vs*: ventral sucker; *gc*: gynecophoral canal; *la*: lesion area; *cp*: ciliated papillae.

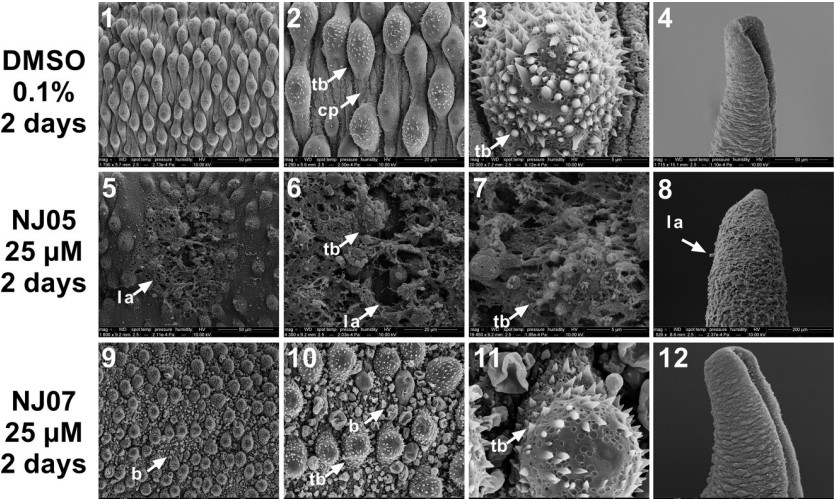

**Fig 6. Scanning electron micrographs of dorsal and posterior regions of *Schistosoma mansoni* adult male worm exposed to NJ05 or NJ07.** Worms were exposed for 2 days to 25 μM NJ05 (panels 5–8), 25 μM NJ07 (panels 9–12) or vehicle (0.1% DMSO) (panels 1–4). **Panels 1 to 4**: Medial (panels 1–3) and posterior (panel 4) regions of control male worm presented normal tegument (Bar = 50 μm; 20 μm; 5 μm; 50 μm). Note that the size scale bar is shown within the black thin line below each image. **Panels 5 to 7**: Enlarged view of dorsal region of adult male worm treated with NJ05 showing tegument lesions and loss of tubercles (Bar = 50 μm; 20 μm; 5 μm; 50 μm); **panel 8**: Posterior region of male adult worm treated with NJ05 showing lesion areas (Bar = 50 μm); **panels 9 to 11**: Dorsal region of male adult worm treated with NJ07 showed bubbles throughout (Bar = 30 μm; 20 μm; 5 μm; 50 μm); **panel 12**: Posterior region of male adult worm treated with NJ07 (Bar = 50 μm). *cp*: ciliated papillae; *b*: blebs; *tb*: tubercles; *la*: lesion area.

the ventral anterior region, focal lesions were observed in the oral and ventral suckers (Fig 5, panels 7 and 8). In the anterior portion of the body exposed to NJ07 the tegument showed no alterations (Fig 5, panels 9–12), and exhibited numerous ciliated papillae similar to the control (Fig 5, panels 1–4).

Severe damage to the dorsal surface of adult male schistosomes was induced by incubation for 2 days with NJ05 (Fig 6, panels 5–8) or NJ07 (Fig 6, panels 9–12). After NJ05 treatment, deterioration of the dorsal tegument was observed at higher magnification (Fig 6, panels 5–7), with the presence of lesion areas and loss of numerous tubercles and spines that covered the whole body of the parasite. In the posterior region, focal lesions along the body became evident (Fig 6, panel 8). Countless bubbles and loss of ciliated papillae were observed in the dorsal region of male worm exposed to NJ07 (Fig 6, panels 9–11); tegument at the male worm posterior region had no structural alterations (Fig 6, panel 12). Similar structural changes were already detected in adult male worms exposed for only 24 h to NJ05 or NJ07, with a more pronounced effect of NJ05 (Fig G in S1 Text).

Control females incubated with vehicle (0.1% DMSO) showed a normal surface (Fig 7, panels 1–4). In females treated with 25 μM NJ05 for 2 days, severe damage to the dorsal tegument was visualized (Fig 7, panels 6–8), and exhibited extensive sloughing that exposed subtegumental tissues. Presence of blisters could be seen throughout the extension of the dorsal region of female exposed to 25 μM NJ07 for 2 days (Fig 7, panels 10–12).

## Assessment by quantitative PCR of the expression of genes involved with vitellaria and female reproduction

Since a strong reduction on oviposition was observed after treatment with NJ series compounds, a set of six genes that are involved in egg formation and participate in cell

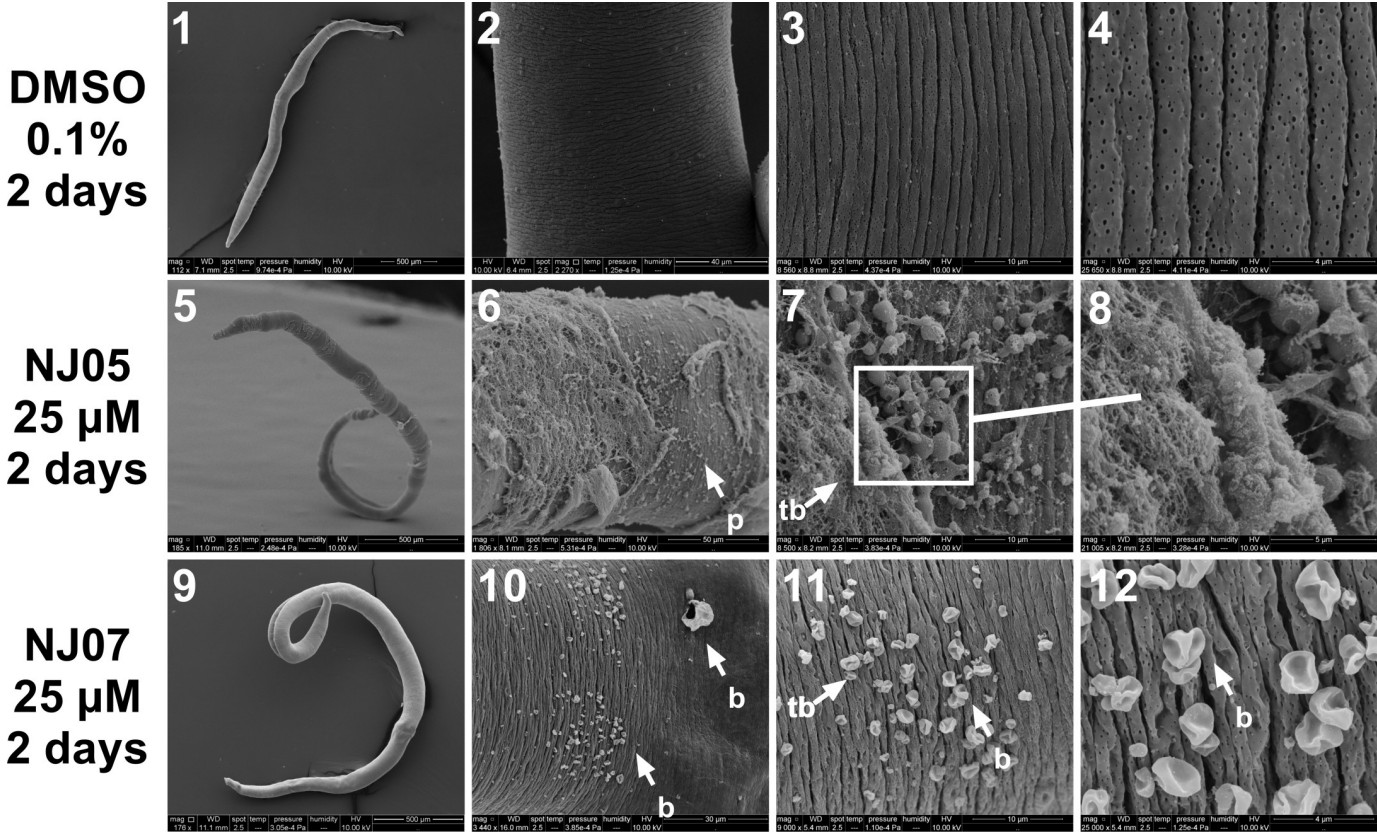

**Fig 7. Damaged dorsal surface of *S. mansoni* adult female worms exposed to NJ05 or NJ07.** Female worms were exposed for 2 days to 25 μM NJ05 (panels 5–8), 25 μM NJ07 (panels 9–12) or to vehicle (0.1% DMSO) (panels 1–4) and were observed by scanning electron microscopy; note that the size scale bar is shown within the black thin line below each image. **panel 1:** Low magnification image of female control worm (Bar = 500 μm). **panels 2 to 4:** Images of female control worms dorsal region, with normal structural appearance (Bar = 40 μm; 10 μm and 4 μm); **panel 5:** Low magnification image of female worm exposed to NJ05 (Bar = 500 μm); **panels 6 to 8:** Middle dorsal region of female presented apparent surface damage with peeling of the tegument membrane (Bar = 50 μm; 10 μm and 5 μm); **panel 9:** Low magnification image of female worm exposed to NJ07 (Bar = 500 μm); **panels 10 to 12:** Dorsal region of female worm presented bubbles along the tegument (Bar = 30 μm; 10 μm and 4 μm). *p*: peeling; *la*: lesion area; *b*: blebs.

differentiation was selected for assessment of their possible change in expression by RT-qPCR. Tested genes involved in egg biosynthesis were all confirmed by RT-qPCR as downregulated in females treated with 25 μM NJ05 for 2 days (Fig 8), with a significant reduction in expression of p14, Tyrosinase 2, p48 and fs800. Two other genes tested by qPCR showed non-statistically-significant changes; they were the Egg Shell Protein (ESP) gene, and the Nanos 2 gene, which is involved in vitelline cell differentiation [41]. It is evident that NJ05 had a direct effect on the expression level of selected genes that are involved with eggshell formation and egg development in *S. mansoni* females [41].

The effect of NJ07 or NJ05 + NJ07 on the expression of the same set of six genes was also tested; worms were treated for 2 days with 25 μM NJ07 or NJ05 + NJ07; the only gene detected by RT-qPCR as significantly 2.8–2.9 X downregulated in females under both treatments was Nanos 2, a gene related to cell differentiation (Fig H in S1 Text). Although oviposition was impaired in both treatment conditions (see Fig 4E and 4F), there was no significant change in expression caused by both treatments among the five tested genes related to eggshell formation and egg development (Fig H in S1 Text), indicating that other genes may have been affected by the treatment.

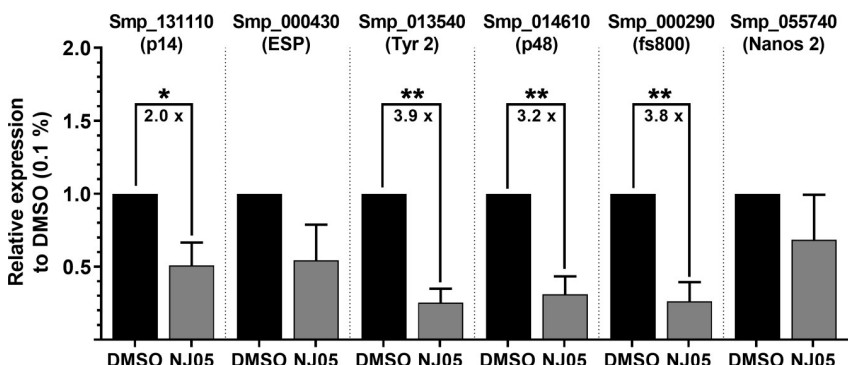

**Fig 8. Vitellaria and female reproduction gene expression inhibition in adult females upon NJ05 treatment.** Adult couples were treated with NJ05 (25 μM) or with vehicle DMSO (0.1%) for 2 days. The parasites were stored in RNAlater (Ambion) for further processing. All parasite couples were separated and only the females had their RNA extracted followed by cDNA synthesis. The genes measured by RT-qPCR were Smp_131110 (p14), Smp_000430 (Egg Shell Protein (ESP)), Smp_013540 (Tyrosinase 2 (Tyr 2)), Smp_014610 (p48), Smp_000290 (fs800), Smp_055740 (Nanos 2). The geometric mean from two reference genes (Smp_090920 and Smp_123610) was used for normalization with the DCT method. The plotted data is retrieved from DDCT analyses in which the DMSO-treated sample is the control. Significant fold-change in gene expression between DMSO and NJ05 treatment is shown by the numbers inside the brackets. Mean ± SEM of four replicates is shown. Student unpaired parametric two-sided t-test was used, and statistically significant differences are represented by the asterisks. $^{*}p \leq 0.05$; $^{**}p \leq 0.01$.

## Discussion

Chemotherapy is the only immediate recourse for minimizing the prevalence of schistosomiasis, and currently it involves predominately the administration of a single drug, praziquantel (PZQ). There is an important limitation of the therapeutic profile of PZQ, as it lacks activity against larval developing stages of the parasite; therefore, retreatment is necessary to kill those parasites that have since matured [21,42]. The present study was performed to evaluate the *in vitro* schistosomicidal effect of the aryl-thiazole derivatives both on adults and on the larval developing stages of the parasite. The following schistosomicidal parameters were analyzed: viability, motility, oviposition, mortality and morphological changes, which are indicators commonly used for evaluating the schistosomicidal effect of new chemical entities [43,44]. In light of the parasite biology and according to the evaluation of these parameters, schistosomicidal activity can be effective essentially in three different ways: causing schistosomula death, thus preventing host infection, inhibiting oviposition that could lead to a decrease in the pathological phenotype of schistosomiasis (hepato- and splenomegaly), and causing adult worm's death [44,45]. All three of the above schistosomicidal effects were present in our *in vitro* assays with NJ compounds, suggesting that further exploration of this class of aryl-thiazole derivatives is warranted.

The aryl-thiazole compounds evaluated in this study are differentiated, in principle, by the substitution pattern of the phenyl ring directly attached to the hydrazine portion (2,4-dimethoxy-phenyl and 3,4,5-trimethoxy-phenyl). Trimethoxylated derivatives (NJ04, NJ05, NJ07 and NJ11) showed a better profile in reducing the viability of *S. mansoni* schistosomula and adult worms compared with the dimethoxylated ones (NJ03, NJ06, NJ08 and NJ012). One important chemical property of methoxy substituent is that its oxygen atom is an acceptor of proton for different types of hydrogen-bonds [46] favoring the biological activity of new compounds by improving the target recognition [47]. Besides, trimethoxy-substitution increases the electronic density, and this could also be related to the higher schistosomicidal activity of this class compared with the dimethoxylated ones. The results corroborate with studies involving other parasites such as *Plasmodium falciparum* [48], *Leishmania braziliensis*

[49] and *Trypanosoma cruzi* [50], evidencing the importance of methoxy group trisubstituted aromatic nuclei for antiparasitic activity.

Additionally, the two aryl-thiazole series are discriminated by the presence of different substituents of the phenyl ring attached to the thiazole nucleus (hydrogen, 4-chloro, 4-bromo, 4-nitro). The best compounds evaluated in this study have 4-nitro (NJ05) and 4-bromo (NJ07). Notably, such substitutions reflect an increase in antiparasitic activity as described in the literature; Papadopoulou *et* al., 2016 [51] described the antitrypanosomal activity of 5-nitro-2-aminothiazole and demonstrated activity against *T. cruzi* amastigotes at nanomolar concentrations and was about 4-fold more potent than BNZ (benznidazole-reference drug). Generally, the biological activity of nitro compounds involves the biotransformation of the nitro group, releasing intermediates that disrupt the redox process. Some of those intermediates can attack enzymes, membranes and DNA, providing the basis for their mode of action, including those reported against parasites [52]. NJ compounds might have caused oxidative stress and the consequent biological activity observed here.

The insertion of halogen atoms, in turn, on the compound structure in the discovery phase of new drugs can lead to increased potency and selectivity of the pharmacological target by affecting the pKa, modulating the conformation, increasing hydrophobic interactions between the ligand and the target, increasing lipophilicity and favoring permeation in biological membranes [53,54]. The increase in cellular permeability of the parasite membrane is an important feature for the action of a target drug; for example, PZQ is supposed to act on the voltage-gated calcium channels of schistosomes [55], but has also been shown to alter membrane fluidity in model membranes [56] and in the *S. mansoni* tegument [57], thus possibly increasing calcium permeability. The effects of para-substituents on biological activity can be explained by the hydrophobicity ($\pi$) and electronic effects ($\sigma$) as stated by Craig, 1971 [58]. Both $NO_2$ and Bromo substituents present positive electronic ($+ \sigma$) and lipophilic ($+ \pi$) effects. This means inferior availability of free electron pairs, which could promote membrane permeation. The phenotypic effects observed over the course of NJ05 treatment on *S. mansoni* couples included reduction in motility, impairment of peristaltic movement and of the ability of their suction suckers to adhere to the bottom of the culture plates, thus generating weak contractions and subsequent paralysis. This may be a consequence of the perturbation in the influx of extracellular calcium, which in eukaryotic cells is a signal for motor events, besides playing an important role in the promotion of apoptosis [59–61]. Further studies are necessary to explore other structural aspects of these compounds besides their possible lipophilicity contribution to membrane permeabilization.

Changes in the general conditions of worms, including adult worms' motility and tegument damage visualized through high resolution scanning electron microscopy were observed after two days treatment with 25 μM NJ05 + NJ07. Motility assay and cytotoxicity evaluation by ATP quantification revealed an increase in the number of dead parasites that was concentration and time dependent following NJ05 + NJ07 treatment. Maximum death of adult worms occurred after three days of treatment at concentrations of 25 or 50 μM NJ05 + NJ07. Noteworthy, in schistosomula 100% death happened in the first two days (25 or 50 μM NJ05 + NJ07). The larvae presented lack of movement, granularity and shape alterations, showing that NJ05 + NJ07 combination was effective against younger forms of the parasite. This is of great importance, since the drugs of choice for treatment of schistosomiasis act on adult worms, having a small action on schistosomula [62].

Detailed microscopic observation showed that NJ05 or NJ07 were capable, separately, of inducing morphological alterations in the outer membrane of all adult worms tested in this work. We found extensive damage to the tegument in both male and female worms after 2 days of exposure to NJ05 or NJ07. Male adult worms of *S. mansoni* treated with NJ05 showed

lesions of tegument and loss of tubercles; the females, in turn, demonstrated severe damage on the tegument surface with the presence of blisters and large peeling of the tegument. Upon NJ07 *in vitro* treatment the dorsal region of male and female adult worms showed bubbles in all their extension. Alterations in the surface ultrastructure of schistosome worms have been investigated by a number of authors in order to evaluate anti-schistosomal drugs [63–68]. Here, the alterations caused by NJ05 or NJ07 indicate that trimethoxy substituent and the presence of $NO_2$ and Br substituents favored damage on male and female adult worms.

The compounds tested here appeared to influence adult females' oviposition, mainly NJ05 and NJ05 + NJ07, since no eggs at all were found in the culture medium after treatment under these two conditions. Eggs obtained from couples treated with NJ07 were smaller than control eggs and probably not viable due to their lack of germinal disks formation. In females treated with NJ05, the expression of genes related to egg biosynthesis including tyrosinase (SmTyr2) [69], and eggshell precursors such as p14 [70] and p48 [71] were found to be decreased in relation to the control.

Also, genes related to cell differentiation and development of the germinative line of metazoans such as Nanos 2 were found to have decreased expression in relation to the control in adult females treated with NJ05, NJ07 or NJ05 + NJ07. On platyhelminths, Nanos 2 is generally found to be expressed in the anterior portion of the ovary [41], where the oogonias are located [72], and it plays an essential role in the proper development, regeneration and maintenance of germ cells in several organisms [73–75].

Overall, the results indicate that thiazoles, especially those trimethoxylated derivatives are antiparasitic agents. These compounds showed substantial schistosomicidal properties against schistosomula and adult *S. mansoni* worms, with a significant reduction in viability, motility, pairing, survival rate, oviposition, severe alterations in the tegument and mortality of worms, and further biological studies are warranted to clarify the mechanisms of schistosomicidal action. It is important to emphasize that NJ05 + NJ07 had an intense activity against the young forms of *S. mansoni in vitro*, a characteristic not encountered in the drugs of choice for treatment of schistosomiasis such as PZQ and oxamniquine [8,76].

Cytotoxic effect for HEK293 human embryonic kidney cell line was higher (EC50 of 46.64, 92.35 and 18.42 µM for NJ05, NJ07 and NJ05+NJ07, respectively) than the cytotoxic effect for HES somatic epithelial cell line (EC50 of 133.53, 145.20 and 29.59 µM for NJ05, NJ07 and NJ05+NJ07, respectively) suggesting that these compounds may be more effective in diminishing viability of embryonic cells that undergo proliferation than in somatic cells. Systematically, the combination between NJ05+NJ07 showed a cytotoxic effect higher than the compounds separately, but it is worth mentioning that the best compound in reducing viability of both schistosomula and adult worms was NJ05. Given that the EC50 for NJ05 in the HEK293 human cell line is in the medium micromolar range (46.64 µM), and in the HES human cell line the EC50 is in the high micromolar range (133.53 µM), this compound could be used as an aryl-thiazole prototype for the development of new schistosomicidal compounds in the future.

## Supporting information

**S1 Text. Supplementary Figs A to H.** Fig A, ATP quantitation using a luminescent assay to assess schistosomula survival under NJ series compounds exposure; Fig B, Effect of NJ series compounds on the viability of *S. mansoni* adult worms; Fig C, Effect of NJ series compounds on the motility of *S. mansoni* adult worms; Fig D, *In vitro* effect of NJ series compounds on pairing of *S. mansoni* adult worm couples; Fig E, *In vitro* effects of NJ series compounds on oviposition of *S. mansoni* adult worms; Fig F, Cytotoxicity evaluation of exposure of two

different human cell lines to NJ series compounds; Fig G, Structural damage was observed by scanning electron microscopy of male worms already at 24 h exposure to NJ05 or NJ07; Fig H, Vitellaria and female reproduction gene expression in adult females upon NJ07 or NJ05+NJ07 treatment.
(PDF)

**S1 Table. List of primers used in qPCR.**
(XLSX)

## Acknowledgments

We thank Beatriz Mauricio and Dr. Carlos Jared, Instituto Butantan, Sao Paulo, for access to the scanning electron microscopy facilities in their laboratories. We acknowledge Dr. Eliana Nakano and Patricia Aoki Miyasato, Instituto Butantan, Sao Paulo, and Dr. Pedro Luiz Silva Pinto, Instituto Adolfo Lutz, Sao Paulo, for maintaining the parasite life cycle.

## Author Contributions

**Conceptualization:** Adriana S. A. Pereira, Sergio Verjovski-Almeida.

**Data curation:** Adriana S. A. Pereira, Gilbert O. Silveira, Sinara M. V. Almeida, Jamerson F. Oliveira.

**Formal analysis:** Adriana S. A. Pereira, Gilbert O. Silveira, Sinara M. V. Almeida, Jamerson F. Oliveira, Maria C. A. Lima.

**Funding acquisition:** Maria C. A. Lima, Sergio Verjovski-Almeida.

**Investigation:** Adriana S. A. Pereira, Murilo S. Amaral.

**Methodology:** Adriana S. A. Pereira, Gilbert O. Silveira, Murilo S. Amaral.

**Project administration:** Sergio Verjovski-Almeida.

**Resources:** Sergio Verjovski-Almeida.

**Supervision:** Sergio Verjovski-Almeida.

**Visualization:** Gilbert O. Silveira.

**Writing – original draft:** Adriana S. A. Pereira, Sergio Verjovski-Almeida.

**Writing – review & editing:** Adriana S. A. Pereira, Gilbert O. Silveira, Sinara M. V. Almeida, Sergio Verjovski-Almeida.

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
