## [Decision Letter · Decision Letter 0]

22 Aug 2019

PONE-D-19-21297

In vitro activity of aryl-thiazole derivatives against Schistosoma mansoni schistosomula and adult worms

PLOS ONE

Dear Professor Sergio Verjovski,

Thank you for submitting your manuscript to PLOS ONE. After careful consideration, we feel that it has merit but does not fully meet PLOS ONE’s publication criteria as it currently stands. Therefore, we invite you to submit a revised version of the manuscript that addresses the points raised during the review process.

The manuscript was reviewed by three experts in the field. Despite the reviewers stating that the present report is in a good shape from a technical point of view, they claim that the manuscript must be revised before it is suitable for publication. In addition, cytotoxicity assays are required.

Although the results are preliminary, particularly due to the lack of in vivo studies, I believe that the study addresses an interesting question. In my view, resubmission of the manuscript is possible, but only if the comments that were raised are fully addressed by providing additional experiments that would be in line with the provided comments. 

We would appreciate receiving your revised manuscript by October 21, 2019. To enhance the reproducibility of your results, we recommend that if applicable you deposit your laboratory protocols in protocols.io, where a protocol can be assigned its own identifier (DOI) such that it can be cited independently in the future. For instructions see: http://journals.plos.org/plosone/s/submission-guidelines#loc-laboratory-protocols

We look forward to receiving your revised manuscript.

Kind regards,

Josué de Moraes, Ph.D.

Academic Editor

PLOS ONE

**Journal Requirements:**

2. We noticed you have some minor occurrence(s) of overlapping text with the following previous publication(s), which needs to be addressed:

https://doi.org/10.1371/journal.pntd.0006873

In your revision ensure you cite all your sources (including your own works), and quote or rephrase any duplicated text outside the Methods section. Further consideration is dependent on these concerns being addressed."""

3. Please amend the subsection category “[FOR JOURNAL STAFF USE ONLY]” for your manuscript. Unfortunately, this is not a valid category. At this time, please choose one or more subsections that best represent the topic(s) of your study.

**Comments to the Author**

1. Is the manuscript technically sound, and do the data support the conclusions?

Reviewer #1: Yes

Reviewer #2: Partly

Reviewer #3: Yes

2. Has the statistical analysis been performed appropriately and rigorously? 

Reviewer #1: Yes

Reviewer #2: Yes

Reviewer #3: Yes

3. Have the authors made all data underlying the findings in their manuscript fully available?

Reviewer #1: Yes

Reviewer #2: No

Reviewer #3: Yes

4. Is the manuscript presented in an intelligible fashion and written in standard English?

Reviewer #1: Yes

Reviewer #2: Yes

Reviewer #3: Yes

5. Review Comments to the Author

Reviewer #1: This manuscript described the schistosomicidal activities of aryl-thiazole derivatives. Some of the data and expression in this manuscript are unclear and should be revised before it can be accepted for publication. My comments are listed below:

1- The manuscript is very difficult to read. Please organize the Material and Methods session. e.g. Title: "Treatment of adult worms with NJ series compounds".

Paragraph: "Schistosomula and adult worms were treated with different concentrations of compounds from the NJ series in culture medium specific to each stage as indicated below..." (page 13). According to the results, adult worms and schistosomula treatments were performed under different conditions (incubation period), please make this information clear in the text;

2- It's very important to remind the necessity of cytotoxicity assay done to prove the absence of toxicity of the aryl-thiazole derivatives, before this manuscript to be accepted for publication;

3- It’s widely studied the correlation between the effect of some drugs on S. mansoni motility through of classical studies. e.g. Hillman, G. R., Senft, A.W.J Pharmacol Exp Ther (1993), 119, 75-86.; Da Silva, S.P., Noel, E. Parasitol Res (1995), 81:543-548; Mendonça Silva, D. L. et al., Parasitology (2004), 129 : 1-10; Rinaldi, G. et al., Int J Parasitol-Drug (2015), 141e148; De Oliveira, R.N., Exp Parasitol (2012), 132,135–143.

In present study, motility and survival of worms were assessed according to the criteria of a nematode (Toxocara canis- ref 30), which are completely different from trematode. Please, why this choice? (Page 14, session “Motility assay");

4- The order and description of the results session is very confusing to the reader;

5- In the Results session: "Phenotypic effects of NJ series compounds of S. mansoni schistosomula" (page 17). The description is very poor and confused;

6- In the Results session:”Fig 2. Synergistic effect of NJ05 + NJ07 on schistosomula viability (A and B) ATP” (page 17). It is extremely premature to describe this result as a synergistic effect of NJ05 + NJ07 compounds, considering that pharmacokinetic and pharmacodynamics mechanisms does not yet been studied. Please review this affirmation with caution;

7- Finally, in the results session "Assessment by quantitative PCR of the expression of genes involved with vitellaria and female reproduction" (page 23).

Whereas: "Tested genes involved in egg biosynthesis were all confirmed by qPCR as downregulated in females treated with 25 μM NJ05 for 2 days (Fig 8), with a significant reduction in expression of p14, Tyrosinase 2, p48 and fs800 ".

Suggestion: It would be very interesting to observe the eggs viability by miracid hatching assay. e.g. Sarvel, A.K. Mem Inst Oswaldo Cruz (2006), 101 (Suppl. I): 289-292.

Major Revision

Reviewer #2: REVIEWER COMMENTS ON THE MANUSCRIPT PONE – D – 19- 21297

The research article entitled “In vitro activity of aryl-thiazole derivatives against Schistosoma mansoni schistosomula and adult worms” reports the synthesis and the antischistosomal characterization of a series of aryl-thiazoles.

The article has distinction when regarding the quantity and quality of the biological assays performed as well as about the synthetic approach and the choice of this chemical class. It lacks, however, the important selectivity information of the compounds since, nowadays, to prove effectivity of a compound are not enough to establish the potential of it to be employed as a lead or even as a hit compound in drug discovery.

No mention about any kind of selectivity assays is present in this study. At least a cytotoxicity assay against some usual cell lines should be conducted, such as VERO, HaCAT or lymphocyte cells.

I also suggest, to a further submission, a careful review of the manuscript in the following aspects:

MATERIAL AND METHODS

Page 6:

Even the synthetic approach being well-known, it would be kind if the authors provided some important aspects of the synthesis, such as the molar proportion between the reagents and the molar excess of sodium acetate employed. Everyone who reads your article could, easily, reproduce the synthesis directly from it. Yet, regarding the reaction media, did the authors employ any solvent that allowed the reflux condition? If yes, please, provide this information along the experimental text.

I also seem as important to include some characterization information about the compounds, such as NMR and IR signals, melting point ranges, etc. The journal public include chemists which would like to be provided with this information. No spectra needed, however. Only the attributions would be enough.

RESULTS

Page 11: Figure 2. Remember to mention that the data did not correspond only to NJ05+NJ07 together but also to NJ05 and NJ07 individually. The Figure 2 title did not reflect the exhibited data.

Page 13: When discussing about decreasing in worm motility, it seems that NJ05+NJ07 together leaded to a higher decrease than NJ05 and NJ07 alone. Similarly to the discussed by the authors about the viability being decreased only by the NJ05, it seemed, to me, that the association decreases in motility was similar to the observed with NJ07 only. Please, review this data.

Page 14: Authors present the results of NJ07 on the worm pairing and oviposition. But, at the end of such discussion, they mention that the other compounds were able to result in impairments of 75 t0 90%. So, these compounds were better in modulate the worm pairing than NJ07 and that is exactly what Figure D from S1 text informs. For instance, NJ03 is better than NJ07 at 50 �M. So, why the authors have chosen NJ07 to discuss along the text and presented the better data in supplementary material?

It that is not the case, please review the phrase construction which induces the reader to understand what I stated.

DISCUSSION

Page 19: Authors constructed a parallel between their findings and the best ways to achieve antischistosomal activity:

“In light of the parasite biology and according to the evaluation of these parameters, schistosomicidal activity can be effective essentially in three different ways: causing schistosomula death, thus preventing host infection, inhibiting oviposition that could lead to a decrease in the pathological phenotype of schistosomiasis (hepato- and splenomegaly), and causing adult worm’s death [44, 45]. All three of the above schistosomicidal effects were present in our in vitro assays with NJ compounds…”

Exactly by the fact that several different actions were observed, SELECTIVITY ASSAYS must be conducted, since it could be possible the existence of different mechanisms of actions which, in turn, could lead to potential side effects. This should be better explored by the authors and is a fundamental step to be accomplished to this article to be accepted.

Page 19: Authors did the following affirmation:

“Trimethoxylated derivatives (NJ04, NJ05, NJ07 and NJ11) showed a better profile in reducing the viability of S. mansoni schistosomula and adult worms. The main electronic change on the molecule caused by tri-substitution is an increase of the electronic density, and this effect significantly increased the schistosomicidal activity, also demonstrated by the synergic effect of compounds NJ05 + NJ07 which differ only by 4-nitro and 4-bromo substituents, respectively.”

It is dangerous to correlates the action to the increase of the electronic density reached with the trimethoxy series, specially when this was the only aspect discussed.

Electronic effects could not be the only ones involved in the compound’s behavior. The 3-OMe group, for instance, could be an important docking point to these compounds in a potential receptor. The right group at the right position could made an enormous difference in compounds activity. This is particularly true for hydrogen bond interaction, which could, for instance, be involved in the 3-MeO group (present only in the trimethoxy series) recognition by the worm’s target. Please, review this statement.

It is true that the trimethoxy aromatic group has already being proved to be important for different antiparasitic activities, but not only by its electronic contributions.

Pages 19-20: Authors composed the hypothesis that “NJ compounds might have caused oxidative stress and the consequent biological activity observed here.” Then they correlate the compound’s action to a possible direct amino acid nitrosylation of SmTGR.

The problem is that nitrosylation processes are related to nitroso group (NO) and not to nitro group itself (NO2).

Nitro group usually suffers reduction up to the hydroxylamine and, further, to amine producing reactive oxygen and nitrogen species which, in turn, destroy DNA and other important cell subunits. It is an unspecific mechanism of action which is also related, sometimes, to unsuitable toxicity profiles.

Nitrosylation, on the other hand, implies the presence of a NO species or a NO donor species which, nitroaromatic systems such as the one studied here, do not produce in vivo.

Please, review this hypothesis.

Pages 20: In the statement “Permeation in biological membranes may have been fundamental for the good activity presented by NJ07.” Be careful!

4-Cl would have been more active if membrane permeation would be such significant aspect since its hydrophobic contribution (� of Hansch = 0.70) is higher than that obtained with the 4-NO2 group (� of Hansch = 0.24) present in NJ05 structure.

You have limited experimental data to comment on the relationship between chemical structure and biological activity by now. Some comments would be interesting but always being careful to highlight the fact that further studies to explore other structural aspects should be performed.

Minor revisions include a careful review of the phrases constructions which are a little bit strange to the formal English.

Reviewer #3: In this manuscript the authors report their work on synthesizing a series of thiazole derivatives and the assessment of the anthelmintic activity of 8 of these compounds in vitro. I feel that this description of their work is very thorough and their study design draws upon all of the best molecular techniques available to synthesize, interrogate and execute these particular class of studies. Their assessments of parasite motility and mortality, egg laying, adult worm pairing and parasite viability clearly demonstrate the antischistosomal activity of their compounds, especially compounds NJ05 and NJ07. The antischistosomal activity of compound NJ05 against juvenile worms shows particular promise given the lack of activity by the gold standard of treatment, praziquantel. The only reason for my suggestion of minor revision include some grammatical issues in the text (e.g. verb tense).

6. PLOS authors have the option to publish the peer review history of their article (what does this mean?). If published, this will include your full peer review and any attached files.

Reviewer #1: No

Reviewer #2: No

Reviewer #3: No

---

## [Author Response · Author response to Decision Letter 0]

19 Oct 2019

Additional Journal Requirements were met.

---

## [Decision Letter · Decision Letter 1]

6 Nov 2019

In vitro activity of aryl-thiazole derivatives against Schistosoma mansoni schistosomula and adult worms

PONE-D-19-21297R1

Dear Dr. Verjovski-Almeida,

We are pleased to inform you that your manuscript has been judged scientifically suitable for publication and will be formally accepted for publication once it complies with all outstanding technical requirements.

With kind regards,

Josué de Moraes, Ph.D.

Academic Editor

PLOS ONE

Reviewers' comments:

Reviewer's Responses to Questions

**Comments to the Author**

1. If the authors have adequately addressed your comments raised in a previous round of review and you feel that this manuscript is now acceptable for publication, you may indicate that here to bypass the “Comments to the Author” section, enter your conflict of interest statement in the “Confidential to Editor” section, and submit your "Accept" recommendation.

Reviewer #1: All comments have been addressed

Reviewer #2: All comments have been addressed

---

## [Editor Report · Acceptance letter]

13 Nov 2019

PONE-D-19-21297R1 

In vitro activity of aryl-thiazole derivatives against Schistosoma mansoni schistosomula and adult worms 

Dear Dr. Verjovski-Almeida:

I am pleased to inform you that your manuscript has been deemed suitable for publication in PLOS ONE. Congratulations! Your manuscript is now with our production department. 

With kind regards,

on behalf of

Dr. Josué de Moraes 

Academic Editor

PLOS ONE